# DEEP LEARNING IS COMPOSITE KERNEL LEARNING

## ABSTRACT

Recent works have connected deep learning and kernel methods. In this paper, we show that architectural choices such as convolutional layers with pooling, skip connections, make deep learning a composite kernel learning method, where the kernel is a (architecture dependent) composition of base kernels: even before training, standard deep networks have in-built structural properties that ensure their success. In particular, we build on the recently developed 'neural path' framework[1] that characterises the role of gates/masks in fully connected deep networks with ReLU activations.

## 1 INTRODUCTION

The success of deep learning is attributed to *feature learning*. The conventional view is that feature learning happens in the hidden layers of a deep network: in the initial layers simple low level features are learnt, and sophisticated high level features are learnt as one proceeds in depth. In this viewpoint, the penultimate layer output is the final hidden feature and the final layer learns a linear model with these hidden features. While this interpretation of feature learning is intuitive, beyond the first couple of layers it is hard make any meaningful interpretation of what happens in the intermediate layers.

Recent works Jacot et al. (2018); Arora et al. (2019); Cao and Gu (2019) have provided a kernel learning interpretation for deep learning by showing that in the limit of infinite width deep learning becomes kernel learning. These works are based on *neural tangents*, wherein, the gradient of the network output with respect to the network parameters known as the *neural tangent features* (NTFs) are considered as the features. Arora et al. (2019) show that at randomised initialisation of weights, the kernel matrix associated with the NTFs, known as the *neural tangent kernel* (NTK) converges to a deterministic matrix and that optimisation and generalisation of infinite width deep neural networks is characterised by this deterministic kernel matrix. Cao and Gu (2019) provided generalisation bounds in terms of the NTK matrix. Arora et al. (2019) also proposed a *pure-kernel* method based on CNTK (NTK of convolutional neural networks, i.e., CNNs) which significantly outperformed the previous state-of-the-art kernel methods. The NTK either as an interpretation or as a method in itself has been very successful. Nevertheless it has some open issues namely i) **non-interpretable:** the kernel is the inner product of gradients and has no physical interpretation, ii) **no feature learning:** the NTFs are random and fixed during training and iii) **performance gap:** finite width CNN outperforms the infinite width CNTK, i.e., NTK does not fully explain the success of deep learning.

Recently, Lakshminarayanan and Singh (2020) developed a *neural path* (NP) framework to provide a kernel interpretation for deep learning that addresses the open issues in the current NTK framework. Here, DNNs with ReLU activations are considered, and the gates (*on/off* state of ReLU) are encoded in the so called *neural path feature* (NPF) and the weights in the network in the so called *neural path value* (NPV). The key findings can be broken into the following steps.

**Step 1:** The NPFs and NPV are decoupled. Gates are treated as masks, which are held in a separate feature network and applied to the main network called the value network. This enables one to study the various kinds of gates (i.e., NPFs), such as random gates (of a randomly initialised network), semi-learnt gates (sampled at an intermediate epoch during training), and learnt gates (sampled from a fully trained network). This addresses the feature learning issue.

**Step 2:** When the gates/masks are decoupled and applied externally it follows that NTK = const × NPK, at random initialisation of weights. For a pair of input examples, NPK is a similarity measure

---

[1]Introduced for the first time in the work of Lakshminarayanan and Singh (2020).

that depends on the size of the sub-network formed by the gates that are active simultaneously for examples. This addresses the interpretability issue.

**Step 3:** CNTK performs better than random gates/masks and gates/masks from fully trained networks perform better than CNTK. This explains the performance gap between CNN and CNTK. It was also observed (on standard datasets) that when learnt gates/masks are used, the weights of the value network can be reset and re-trained from scratch without significant loss of performance.

## 1.1 CONTRIBUTIONS IN THIS WORK

We attribute the success of deep learning to the following two key ingredients: (i) a composite kernel with gates as fundamental building blocks and (ii) allowing the gates to learn/adapt during training. Formally, we extend the NP framework of Lakshminarayanan and Singh (2020) as explained below.

• **Composite Kernel:** The NPK matrix has a composite structure (architecture dependent).

1. *Fully-Connected* networks: $H^{\mathrm{fc}}$ is the *Hadamard* product of the input data Gram matrix, and the kernel matrices corresponding to the binary gating features of the individual layers.

2. *Residual* networks (ResNets) with *skip* connections: $H^{\mathrm{res}}$ assumes a sum of products form. In particular, consider a ResNet with $(b + 2)$ blocks and $b$ skip connections. Within this ResNet there are $i = 1, \ldots, 2^b$ possible dense networks, and then $H^{\mathrm{res}} = \sum_{i=1}^{2^b} C_i H_i^{\mathrm{fc}}$, where $C_i > 0$ are positive constants based on *normalisation* layers.

3. *Convolutional* neural networks (CNN) with *pooling*: $H^{\mathrm{cnn}}$ is rotation invariant.

• **Gate Learning:** We show that learnt gates perform better than random gates. Starting with the setup of Lakshminarayanan and Singh (2020), we build *combinatorially* many models by,

1. permuting the order of the layers when we apply them as external masks,

2. having two types of modes based on input provided to the value network namely i) 'standard': input is the actual image and ii) 'all-ones': input is a tensor with all entries as '1'.

We observe in our experiments that the performance is robust to such combinatorial variations.

**Message:** This work along with that of Lakshminarayanan and Singh (2020) provides a paradigm shift in understanding deep learning. Here, gates play a central role. Each gate is related to a hyperplane, and gates together form layer level binary features whose kernels are the base kernels. Laying out these binary features depth-wise gives rise to a product of the base kernels. The skip connections gives a 'sum of product' structure, and convolution with pooling gives rotation invariance.

**Organisation:** Section 2 describes the network architectures namely fully-connected, convolutional and residual, which we take up for theoretical analysis. Section 3 extends the neural path framework to CNN and ResNet. Section 4 explains the *composite* kernel. Section 5 connects the NTK and NPK for CNN and ResNet. Section 6 consists of numerical experiments.

## 2 ARCHITECTURES: FULLY CONNECTED, CONVOLUTIONAL AND RESIDUAL

In this section, we present the three architectures that we take up for theoretical analysis. These are i) fully connected (FC or FC-DNN), ii) convolutional (CNN) and iii) residual (ResNets). In what follows, $[n]$ is the set $\{1, \ldots, n\}$, and the dataset is given by $(x_s, y_s)_{s=1}^n \in \mathbb{R}^{d_{\mathrm{in}}} \times \mathbb{R}$.

**Fully Connected:** We consider fully connected networks with width '$w$' and depth '$d$'.

**CNN:** We consider a 1-dimensional convolutional neural network with circular convolutions (see Table 2), with $d_{\mathrm{cv}}$ convolutional layers ($l = 1, \ldots, d_{\mathrm{cv}}$), followed by a *global-average/max-pooling* layer ($l = d_{\mathrm{cv}} + 1$) and $d_{\mathrm{fc}}$ ($l = d_{\mathrm{cv}} + 2, \ldots, d_{\mathrm{cv}} + d_{\mathrm{fc}} + 1$) FC layers. The convolutional window size is $w_{\mathrm{cv}} < d_{\mathrm{in}}$, the number of filters per convolutional layer is $w$, and the width of the FC is also $w$.

**Definition 2.1** (Circular Convolution). *For $x \in \mathbb{R}^{d_{in}}$, $i \in [d_{in}]$ and $r \in \{0, \ldots, d_{in} - 1\}$, define :*

*(i) $i \oplus r = i + r$, for $i + r \leq d_{in}$ and $i \oplus r = i + r - d_{in}$, for $i + r > d_{in}$.*

*(ii) $rot(x, r)(i) = x(i \oplus r), i \in [d_{in}]$.*

| Input Layer | : | $z_{x,\Theta}(\cdot, 0)$ | $=$ | $x$ |
|---|---|---|---|---|
| Pre-activation | : | $q_{x,\Theta}(i_{\text{out}}, l)$ | $=$ | $\sum_{i_{\text{in}}} \Theta(i_{\text{in}}, i_{\text{out}}, l) \cdot z_{x,\Theta}(i_{\text{in}}, l-1)$ |
| Gating Value | : | $G_{x,\Theta}(i_{\text{out}}, l)$ | $=$ | $\mathbf{1}_{\{q_{x,\Theta}(i_{\text{out}}, l) > 0\}}$ |
| Hidden Unit Output | : | $z_{x,\Theta}(i_{\text{out}}, l)$ | $=$ | $q_{x,\Theta}(i_{\text{out}}, l) \cdot G_{x,\Theta}(i_{\text{out}}, l)$ |
| Final Output | : | $\hat{y}_{\Theta}(x)$ | $=$ | $\sum_{i_{\text{in}}} \Theta(i_{\text{in}}, i_{\text{out}}, d) \cdot z_{x,\Theta}(i_{\text{in}}, d-1)$ |

Table 1: Information flow in a FC-DNN with ReLU. Here, '$q$'s are pre-activation inputs, '$z$'s are output of the hidden layers, '$G$'s are the gating values. $l \in [d-1]$ is the index of the layer, $i_{\text{out}}$ and $i_{\text{in}}$ are indices of nodes in the current and previous layer respectively.

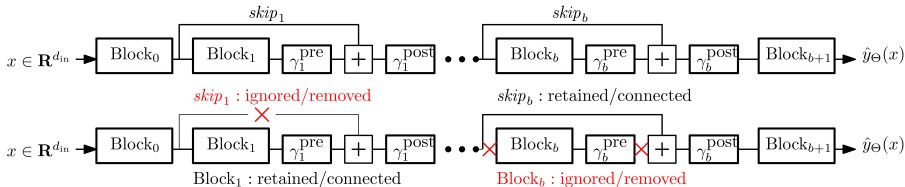

Figure 1: ResNet Architecture is shown in the top. Process of obtaining a sub-FC-DNN by ignoring skip (retaining block) or retaining skip (ignoring block) is shown in the bottom.

*(iii)* $q_{x,\Theta}(i_{fout}, i_{out}, l) = \sum_{i_{cv}, i_{in}} \Theta(i_{cv}, i_{in}, i_{out}, l) \cdot z_{x,\Theta}(i_{fout} \oplus (i_{cv} - 1), i_{in}, l-1)$, *where $i_{in}/i_{out}$ are the indices (taking values in $[w]$) of the input/output filters. $i_{cv}$ denotes the indices of the convolutional window (taking values in $[w_{cv}]$) between input and output filters $i_{in}$ and $i_{out}$. $i_{fout}$ denotes the indices (taking values in $[d_{in}]$, the dimension of input features) of individual nodes in a given output filter.*

**Definition 2.2** (Pooling). *Let $G^{pool}_{x,\Theta}(i_{fout}, i_{out}, d_{cv} + 1)$ denote the pooling mask, then we have*

$$z_{x,\Theta}(i_{out}, d_{cv} + 1) = \sum_{i_{fout}} z_{x,\Theta}(i_{fout}, i_{out}, d_{cv}) \cdot G^{pool}_{x,\Theta}(i_{fout}, i_{out}, d_{cv} + 1),$$

*where in the case of* global-average-pooling $G^{pool}_{x,\Theta}(i_{fout}, i_{out}, d_{cv} + 1) = \frac{1}{d_{in}}, \forall i_{out} \in [w], i_{fout} \in [d_{in}]$, *and in the case of* max-pooling*, for a given* $i_{out} \in [w]$, $G^{pool}_{x,\Theta}(i_{max}, i_{out}, d_{cv} + 1) = 1$ *where* $i_{max} = \arg\max_{i_{fout}} z_{x,\Theta}(i_{fout}, i_{out}, d_{cv})$, *and* $G^{pool}_{x,\Theta}(i_{fout}, i_{out}, d_{cv} + 1) = 0, \forall i_{fout} \neq i_{max}$.

**ResNet:** We consider ResNets with '$(b+2)$' blocks and '$b$' skip connections between the blocks (Figure 1). Each block is a FC-DNN of depth '$d_{\text{blk}}$' and width '$w$'. Here, $\gamma_i^{\text{pre}}, \gamma_i^{\text{post}}, i \in [b]$ are normalisation variables.

**Definition 2.3** (Sub FC-DNNs). *Let $2^{[b]}$ denote the power set of $[b]$ and let $\mathcal{J} \in 2^{[b]}$ denote any subset of $[b]$. Define the '$\mathcal{J}^{th}$' sub-FC-DNN of the ResNet to be the fully connected network obtained by ignoring/removing (see Figure 1) the skip connections $skip_j, \forall j \in \mathcal{J}$ (see Figure 1).*

## 3 NEURAL PATH FRAMEWORK

In this section, we extend the *neural path* framework developed by LS2020, to CNN and ResNet architectures described in the previous section. The neural path framework exploits the gating property of ReLU activation, which can be thought of as gate/mask that blocks/allows its pre-activation input depending on its $0/1$ state ( 0 if pre-activation is negative and 1 if pre-activation is positive). The key idea here is to break a DNN (with ReLU) into paths, and express its output as a summation of the contribution of the paths. The contribution of a path is the product of the signal in its input node, the weights in the path and the gates in the path. For a DNN with $P$ paths, for an input $x \in \mathbb{R}^{d_{\text{in}}}$, the gating information is encoded in a novel *neural path feature* (NPF), $\phi_{x,\Theta} \in \mathbb{R}^P$ and a novel *neural path value* (NPV), $v_{\Theta} \in \mathbb{R}^P$ encodes the weights. The output of the DNN is then the inner product of the NPFs and NPVs, i.e., $\hat{y}_{\Theta}(x_s) = \langle \phi_{x_s,\Theta}, v_{\Theta} \rangle$ (Proposition 3.4).

**Definition 3.1.** *A path starts from an input node, passes through weights, hidden nodes, and normalisation constants and ends at the output node.*

**Proposition 3.1.** *The total number of paths in FC-DNN, CNN and ResNet are respectively given by $P^{fc} = d_{in}w^{(d-1)}$, $P^{cnn} = d_{in}(w_{cv}w)^{d_{cv}}w^{(d_{fc}-1)}$ and $P^{res} = d_{in} \cdot \sum_{i=0}^{b} \binom{b}{i} w^{(i+2)d_{blk}-1}$.*

**Notation 3.1** (Index Maps)**.** *The ranges of index maps $\mathcal{I}_l^f$, $\mathcal{I}_l^{cv}$, $\mathcal{I}_l$ are $[d_{in}]$, $[w_{cv}]$ and $[w]$ respectively. The index maps are used to identify the nodes through which a path $p$ passes. Further, let $\mathcal{I}^{\mathcal{J}}(p)\colon [P^{res}] \to 2^{[b]}$ specify the indices of the skip connections ignored in path $p$. Also, we follow the convention that weights and gating values of layers corresponding to blocks skipped are $1$.*

**Definition 3.2** (Path Activity)**.** *The product of the gating values in a path $p$ is its 'activity' denoted by $A_\Theta(x, p)$. We define:*

*(a) $A_\Theta(x, p) = \Pi_{l=1}^{d-1} G_{x,\Theta}(\mathcal{I}_l(p), l)$, for FC-DNN and ResNet.*

*(b) $A_\Theta(x, p) = \Pi_{l=1}^{d_{cv}+1} G_{x,\Theta}(\mathcal{I}_l^f(p), \mathcal{I}_l(p), l) \cdot \Pi_{l=d_{cv}+2}^{d_{cv}+d_{fc}+1} G_{x,\Theta}(\mathcal{I}_l(p), l)$, for CNN.*

*In CNN, the pooling layer is accounted by letting $G = G^{pool}$ for $l = d_{cv} + 1$.*

**Definition 3.3** (Bundle Paths of Sharing Weights)**.** *Let $\hat{P}^{cnn} = \frac{P^{cnn}}{d_{in}}$, and $\{B_1, \ldots, B_{\hat{P}^{cnn}}\}$ be a collection of sets such that $\forall i, j \in [\hat{P}^{cnn}], i \neq j$ we have $B_i \cap B_j = \emptyset$ and $\cup_{i=1}^{\hat{P}^{cnn}} B_i = [P^{cnn}]$. Further, if paths $p, p' \in B_i$, then $\mathcal{I}_l^{cv}(p) = \mathcal{I}_l^{cv}(p'), \forall l = 1, \ldots, d_{cv}$ and $\mathcal{I}_l(p) = \mathcal{I}_l(p'), \forall l = 0, \ldots, d_{cv}$.*

**Proposition 3.2.** *There are exactly $d_{in}$ paths in a bundle.*

**Definition 3.4** (Normalisation Factor)**.** *Define $\Gamma(\mathcal{J}) \stackrel{\Delta}{=} \underset{j \in J}{\Pi} \gamma_j^{pre} \cdot \underset{j' \in [b]}{\Pi} \gamma_{j'}^{post}$*

Weight sharing is shown in the the cartoon in Figure 2, which shows a CNN with $d_{\text{in}} = 3$, $w = 1$, $w_{\text{cv}} = 2$, $d_{\text{cv}} = 3$, $d_{\text{fc}} = 0$. Here, the red coloured paths all share the same weights $\Theta(1, 1, 1, l), l = 1, 2, 3$ and the blue coloured paths all share the same weights given by $\Theta(2, 1, 1, l), l = 1, 2, 3$.

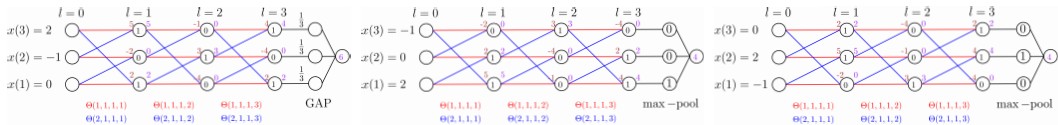

Figure 2: Shows weight sharing and rotational symmetry of internal variables and the output after pooling in a CNN. Left most cartoon uses a GAP layer, and the other two cartoons use $\max$-pooling. Circles are nodes and the $1/0$ in the nodes indicate the gating. Pre-activations/node output are shown in brown/purple.

**Definition 3.5** (Neural Path Value)**.** *The product of the weights and normalisation factors in a path $p$ is its 'value'. The value of a path bundle is the value of any path in that bundle. The path/bundle values are denoted by $v_\Theta(p)/v_\Theta(B_{\hat{p}})$ and are defined as follows:*

*(a) $v_\Theta(p) = \Pi_{l=1}^d \Theta(\mathcal{I}_{l-1}(p), \mathcal{I}_l(p), l)$.*

*(b) $v_\Theta(B_{\hat{p}}) = \Pi_{l=1}^{d_{cv}} \Theta(\mathcal{I}_l^{cv}(p), \mathcal{I}_{l-1}(p), \mathcal{I}_l(p), l) \cdot \Pi_{l=d_{cv}+2}^{d_{cv}+d_{fc}+1} \Theta(\mathcal{I}_{l-1}(p), \mathcal{I}_l(p), l)$, for any $p \in B_{\hat{p}}$.*

*(c) $v_\Theta(p) = \Pi_{l=1}^d \Theta(\mathcal{I}_{l-1}(p), \mathcal{I}_l(p), l) \cdot \Gamma(\mathcal{I}^{\mathcal{J}}(p))$.*

*The neural path value is defined as $v_\Theta \stackrel{\Delta}{=} (v_\Theta(p), p \in [P^{fc}]) \in \mathbb{R}^{P^{fc}}$, $v_\Theta \stackrel{\Delta}{=} (v_\Theta(B_{\hat{p}}), \hat{p} \in [\hat{P}^{cnn}]) \in \mathbb{R}^{\hat{P}^{cnn}}$, and $v_\Theta \stackrel{\Delta}{=} (v_\Theta(p), p \in [P^{res}]) \in \mathbb{R}^{P^{res}}$ for FC-DNN, CNN and ResNet respectively.*

**Proposition 3.3** (Rotational Invariance)**.** *Internal variables in the convolutional layers are circularly symmetric, i.e., for $r \in \{0, \ldots, d_{in} - 1\}$ it follows that (i) $z_{rot(x,r),\Theta}(i_{fout}, \cdot, \cdot) = z_{x,\Theta}(i_{fout} \oplus r, \cdot, \cdot)$, (ii) $q_{rot(x,r),\Theta}(i_{fout}, \cdot, \cdot) = q_{x,\Theta}(i_{fout} \oplus r, \cdot, \cdot)$ and (iii) $G_{rot(x,r),\Theta}(i_{fout}, \cdot, \cdot) = G_{x,\Theta}(i_{fout} \oplus r, \cdot, \cdot)$.*

**Definition 3.6.** *The neural path feature (NPF) corresponding to a path $p$ is given by*

*(a) $\phi_{x,\Theta}(p) \stackrel{\Delta}{=} x(\mathcal{I}_0^f(p)) A_\Theta(x_s, p)$ for FC-DNN and ResNet.*

*(b) $\phi_{x,\Theta}(\hat{p}) \stackrel{\Delta}{=} \sum_{\hat{p} \in B_{\hat{p}}} x(\mathcal{I}_0^f(p)) A_\Theta(x, p)$ for CNN.*

*The NPF is defined as $\phi_{x,\Theta} \stackrel{\Delta}{=} (\phi_{x,\Theta}(p), p \in [P^{fc}]) \in \mathbb{R}^{P^{fc}}$, $\phi_{x,\Theta} \stackrel{\Delta}{=} (\phi_{x,\Theta}(B_{\hat{p}}), \hat{p} \in [\hat{P}^{cnn}]) \in \mathbb{R}^{\hat{P}^{cnn}}$, and $\phi_{x,\Theta} \stackrel{\Delta}{=} (\phi_{x,\Theta}(p), p \in [P^{res}]) \in \mathbb{R}^{P^{res}}$ for FC-DNN, CNN and ResNet respectively.*

**Proposition 3.4** (Output=⟨NPF,NPV⟩)**.** *The output of the network can be written as an inner product of the NPF and NPV, i.e., $\hat{y}_\Theta(x) = \langle \phi_{x,\Theta}, v_\Theta \rangle$.*

## 4 NEURAL PATH KERNEL: COMPOSITE KERNEL BASED ON SUB-NETWORKS

In this section, we will discuss the properties of *neural path kernel* (NPK) associated with the NPFs defined in Section 3. Recall that a co-ordinate of NPF can be non-zero only if the corresponding path is active. Consequently, the NPK for a pair of input examples is a similarity measure that depends on the number of paths that are active for both examples. Such common active paths are captured in a quantity denoted by $\Lambda$ (Definition 4.2). The number of active paths are in turn dependent on the number of active gates in each layer, a fact that endows the NPK with a hierarchical/composite structure. Gates are the basic building blocks, and the gates in a layer for a $w$-dimensional binary feature whose kernels are the base kernels. When the layers are laid out depth-wise, we obtain a product of the base kernels. When skip connections are added, we obtain a sum of products of base kernels. And presence of convolution with pooling provides rotational invariance.

**Definition 4.1.** *Define the NPK matrix to be* $H_\Theta \triangleq \Phi_\Theta^\top \Phi_\Theta$, *where* $\Phi_\Theta = (\phi_{x_1,\Theta}, \ldots, \phi_{x_n,\Theta}) \in \mathbb{R}^{P \times n}$ *is the NPF matrix.*

**Definition 4.2.** *Define* $\Lambda_\Theta(i, x, x_{s'}) \triangleq |\{p \in [P]: \mathcal{I}_0(p) = i, A_\Theta(x_s, p) = A_\Theta(x_{s'}, p) = 1\}|$ *to be total number of 'active' paths for both* $x_s$ *and* $x_{s'}$ *that pass through input node* $i$.

**Definition 4.3** (Layer-wise Kernel). *Let* $G_{x,\Theta}(\cdot, l) \in \mathbb{R}^w$ *be* $w$-*dimensional feature of the gating values in layer* $l$ *for input* $x \in \mathbb{R}^{d_{in}}$. *Define layer-wise kernels:*

$$H_{l,\Theta}^{lyr}(s, s') \triangleq \langle G_{x_s,\Theta}(\cdot, l) G_{x_{s'},\Theta}(\cdot, l) \rangle$$

**Lemma 4.1** (Product Kernel). *Let* $H^{fc}$ *denote the NPK of a FC-DNN, and for* $D \in \mathbb{R}^{d_{in} \times d_{in}}$ *be a diagonal matrix with strictly positive entries, and* $u, u' \in \mathbb{R}^{d_{in}}$ *let* $\langle u, u' \rangle_D = \sum_{i=1}^{d_{in}} D(i)u(i)u'(i)$.

$$H_\Theta^{fc}(s, s') = \langle x_s, x_{s'} \rangle_{\Lambda(\cdot, x_s, x_{s'})} = \langle x_s, x_s' \rangle \Pi_{l=1}^{d-1} H_{l,\Theta}^{lyr}(s, s')$$

**Lemma 4.2** (Sum of Product Kernel). *Let* $H_\Theta^{res}$ *be the NPK of the ResNet, and* $H_\Theta^{\mathcal{J}}$ *be the NPK of the sub-FC-DNN within the ResNet obtained by ignoring those skip connections in the set* $\mathcal{J}$. *Then,*

$$H_\Theta^{res} = \sum_{\mathcal{J} \in 2^{[b]}} H_\Theta^{\mathcal{J}}$$

**Lemma 4.3** (Rotational Invariant Kernel). *Let* $H_\Theta^{cnv}$ *denote the NPK of a CNN, then*

$$H_\Theta^{cnv}(s, s') = \sum_{r=0}^{d_{in}-1} \langle x_s, rot(x_{s'}, r) \rangle_{\Lambda(\cdot, x_s, rot(x_{s'}, r))} = \sum_{r=0}^{d_{in}-1} \langle rot(x_s, r), x_{s'} \rangle_{\Lambda(\cdot, rot(x_s, r), x_{s'})}$$

## 5 MAIN THEORETICAL RESULT

In this section, we proceed with the final step in extending the neural path theory to CNN and ResNet. As with LS2020, we first describe the deep gated network (DGN) setup that decouples the NPFs and NPV, and follow it up with the main result that connects the NPK and the NTK in the DGN setting.

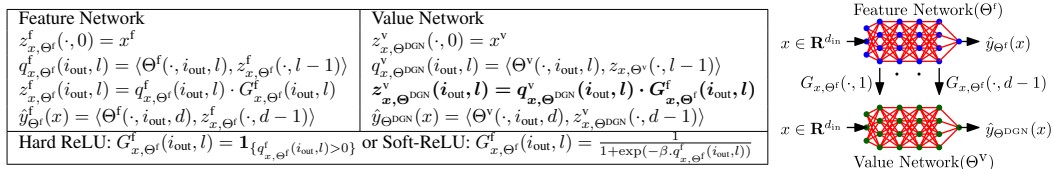

Figure 3: Shows a deep gated network (DGN). The soft-ReLU enables gradient flow into the feature network.

**DGN** set up was introduced by LS2020 to analytically characterise the role played by the gates in a 'standalone' manner. The DGN has two networks namely the *feature network* parameterised by $\Theta^f \in \mathbb{R}^{d_{net}^f}$ which holds the NPFs (i.e., the gating information) and a *value network* parameterised by $\Theta^v \in \mathbb{R}^{d_{net}^v}$ which holds the NPV. The combined parameterisation is denoted by $\Theta^{DGN} = (\Theta^f, \Theta^v) \in \mathbb{R}^{d_{net}^f + d_{net}^v}$. Thus the learning problem in the DGN is $\hat{y}_{\Theta^{DGN}}(x) = \langle \phi_{x,\Theta^f}, v_{\Theta^v} \rangle$.

**Definition 5.1.** *The DGN has **4 regimes** namely decoupled learning (DL), fixed learnt (FL), fixed random-dependent initialisation (FR-DI) and fixed random-independent initialisation (FR-II). In all the regimes $\hat{y}_{\Theta^{DGN}}$ is the output, and $\Theta_0^v$ is always initialised at random and is trainable. However, the regimes differ based on i) trainability of $\Theta^f$, ii) initialisation $\Theta_0^f$ as described below.*

   DL     :   $\Theta^f$ *is trainable, and $\Theta_0^f$ and $\Theta_0^v$ are random and statistically independent, $\beta > 0$.*

   FL     :   $\Theta^f$ *is non-trainable, and $\Theta_0^f$ is pre-trained; $\Theta_0^v$ is statistically independent of $\Theta_0^f$.*

   FR-II  :   $\Theta^f$ *is non-trainable, and $\Theta_0^f$ and $\Theta_0^v$ are random and statistically independent.*

   FR-DI :   $\Theta^f$ *is non-trainable, and $\Theta_0^f = \Theta_0^v$.*

**DGN Regimes:** The flexibility in a DGN is that a) $\Theta^f$ can be trainable/non-trainable and b) $\Theta_0^f$ can be random or pre-trained using $\hat{y}_{\Theta^f}$ as the output (Definition 5.1). By using the DGN setup we can study the role of gates by comparing (a) learnable (DL) vs fixed gates (FL, FR-DI, FR-II), (b) random (FR-DI, FR-II) vs learnt gates (FL) and (c) dependent (FR-DI) vs independent initialisations (FR-II). In the DL regime 'soft-ReLU' is chosen to enable gradient flow through the feature network.

**Proposition 5.1.** *Let $K_{\Theta^{DGN}}$ be the NTK matrix of the DGN, then $K_{\Theta^{DGN}} = K_{\Theta^{DGN}}^v + K_{\Theta^{DGN}}^f$, with*

| Overall NTK | $K_{\Theta^{DGN}}(s, s') = \langle \psi_{x_s, \Theta^{DGN}}, \psi_{x_{s'}, \Theta^{DGN}} \rangle$, where $\psi_{x, \Theta^{DGN}} = \nabla_{\Theta^{DGN}} \hat{y}_{\Theta^{DGN}}(x) \in \mathbb{R}^{d_{net}}$ |
|---|---|
| Feature NTK | $K_{\Theta^{DGN}}^v(s, s') = \langle \psi_{x_s, \Theta^{DGN}}^v, \psi_{x_{s'}, \Theta^{DGN}}^v \rangle$, where $\psi_{x, \Theta^{DGN}}^v = \nabla_{\Theta^v} \hat{y}_{\Theta^{DGN}}(x) \in \mathbb{R}^{d_{net}^v}$ |
| Value NTK | $K_{\Theta^{DGN}}^f(s, s') = \langle \psi_{x_s, \Theta^{DGN}}^f, \psi_{x_{s'}, \Theta^{DGN}}^f \rangle$, where $\psi_{x, \Theta^{DGN}}^f = \nabla_{\Theta^f} \hat{y}_{\Theta^{DGN}}(x) \in \mathbb{R}^{d_{net}^f}$ |

**Remark:** There are two separate NTKs, each one corresponding to feature and value networks respectively. In the case of fixed regimes, $K^f = 0$.

**Theorem 5.1.** *(i) $\Theta_0^v$ is statistically independent of $\Theta_0^f$ (ii) $\Theta_0^v$ are i.i.d symmetric Bernoulli over $\{-\sigma, +\sigma\}$. Let $\sigma_{fc} = \frac{c_{scale}}{\sqrt{w}}$ and $\sigma_{cv} = \frac{c_{scale}}{\sqrt{w w_{cv}}}$ for FC and convolutional layers. As $w \to \infty$, we have:*

*(ii) $K_{\Theta_0^{DGN}}^v \to \beta_{fc} H_{\Theta_0^f}$, $\beta_{fc} = d\sigma_{fc}^{2(d-1)}$ for FC-DNN,*

*(ii) $K_{\Theta_0^{DGN}}^v \to \beta_{cv} H_{\Theta_0^f}$, $\beta_{cv} = \frac{1}{d_{in}^2} \left( d_{cv} \sigma_{cv}^{2(d_{cv}-1)} \sigma_{fc}^{2d_{fc}} + d_{fc} \sigma_{cv}^{2d_{cv}} \sigma_{fc}^{2(d_{fc}-1)} \right)$ for CNN with GAP,*

*(iii) $K_{\Theta_0^{DGN}}^v \to \sum_{\mathcal{J} \in 2^{[b]}} \beta_{rs}^{\mathcal{J}} H_{\Theta_0^f}^{\mathcal{J}}$, $\beta_{rs}^{\mathcal{J}} = (|\mathcal{J}| + 2) d_{blk} \sigma_{fc}^{2\left((|\mathcal{J}|+2)d_{blk}-1\right)} \Gamma(\mathcal{J})^2$ for ResNet.*

• **$\beta_{fc}, \beta_{cv}, \beta_{rs}$ :** The simplest of all is $\beta_{fc} = d\sigma_{fc}^{2(d-1)}$, where $d$ is due the fact that there are $d$ weights in a path and in the exponent of $\sigma_{fc}$, factor $(d-1)$ arises because the gradient of a particular weight is product of all the weights in the path excluding the said weight itself, and the factor of 2 is due to the fact that NTK is an inner product of two gradients. $\beta_{cv}$ is similar to $\beta_{fc}$ with separate bookkeeping for the convolutional and FC layers, and $\frac{1}{d_{in}^2}$ is due to the GAP layer. In $\beta_{rs}$, the $\beta_{fc}$ for all the sub-FC-DNNs within the ResNet are scaled by the corresponding normalisation factors and summed.

• **Decoupling** In a DNN with ReLU (and FR-DI regime of DGN), NPV and NPF are not statistically independent at initialisation, i.e., Theorem 5.1 does not hold. However, the current state-of-the-art analysis Jacot et al. (2018); Arora et al. (2019); Cao and Gu (2019) is in the *infinite width* ($w \to \infty$) regime, wherein, the change in activations during training is only of the order $\sqrt{\frac{1}{w}}$, which goes to 0 as $w \to \infty$. Hence, though assumption in Theorem 5.1 may not hold exactly, it is *not a strong assumption* to fix the NPFs for the purpose of analysis. Once the NPFs are fixed, it only natural to statistically decouple the NPV from fixed NPFs (Theorem 5.1 hold in FR-II, FL and DL regimes).

• **Gates are Key:** In simple terms, Theorem 5.1 says that if the gates/masks are known, then the weights are expendable, a fact which we also verify in our extensive experiments.

## 6 NUMERICAL EXPERIMENTS

We now show via experiments that gates indeed play a central role in deep learning. For this we use the DGN setup (Figure 4) to create models in the 4 regimes namely DL, FL, FR-II and FR-DI. In

each of the 4 regimes, we create combinatorially many models via a) permutation of the layers when the copied from the feature to the value network, and b) setting the input to the value network to $\mathbf{1}$ (in training and testing), i.e., a tensor with all its entries to be 1. We observe that in all the 4 regimes, the models are robust to the combinatorial variations.

**Setup:** Datasets are MNIST and CIFAR-10. For CIFAR-10, we use Figure 4 with $3 \times 3$ windows and 128 filters in each layer. For MNIST, we use FC instead of the convolutional layers. All the FC-DNNs and CNNs are trained with '*Adam*' [10] (step-size $= 3 \cdot 10^{-4}$, batch size $= 32$). A ResNet called *DavidNet* [12] was trained with *SGD* ( step-size $= 0.5$, batch size $= 256$). We use $\beta = 10$.

**Reporting of Statistics:** The results are summarised in Figure 4. For FC-DNN and CNN, in each of the 4 regimes, we train $48 = 2(x^{\mathrm{v}} = x/x^{\mathrm{v}} = \mathbf{1}) \times 24$(layer permutations) models. Each of these models are trained to almost 100% accuracy and the test performance is taken to be the best obtained in a given run. Each of the 48 models is run only once. For the ResNet, we train only two model for each of the 4 regimes ( without permuting the layers, but with image as well as 'all-ones' input variation) and here each mode is run 5 times.

• **Result Discussion:** Recall that in regimes FR-II and FR-DI the gates are fixed and random, and only $\Theta^{\mathrm{v}}$ are trained. In DL regime, both $\Theta^{\mathrm{f}}$ and $\Theta^{\mathrm{v}}$ are trained, and FL regime $\Theta^{\mathrm{f}}$ is pre-trained and fixed, and only $\Theta^{\mathrm{v}}$ is trained. In the following discussion, we compare the performance of the models in various regimes, along with the performance of CNTK of Arora et al. (2019) (77.43% in CIFAR-10) and the performance of standard DNN with ReLU. The main observations are listed below (those by Lakshminarayanan and Singh (2020) are also revisited for the sake of completeness).

1. *Decoupling:* There is no performance difference between FR-II and FR-DI.Further, decoupled learning of gates (DL) performs significantly better than fixed random gates (FR), and the gap between standard DNN with ReLU and DL is less than 3%. This marginal performance loss seems to be worthy trade off for fundamental insights of Theorem 5.1 under the decoupling assumption.

2. *Recovery:* The fixed learnt regime (FL) shows that using the gates of a pre-trained ReLU network, performance can be recovered by training the NPV. Also, by interpreting the input dependent component of a model to be the features and the input independent component to be the weights, it makes sense to look at the gates/NPFs as the hidden features and NPV as the weights.

3. *Random Gates:* FR-II does perform well in all the experiments (note that for a 10-class problem, a random classifier would achieve only 10% test accuracy). Given the observation that the gates are the true features, and the fact that is no learning in the gates in the fixed regime, and the performance of fixed random gates can be purely attributed to the in-built structure.

4. *Gate Learning:* We group the models into three sets where $S_1 = \{$ ReLU, FL , DL$\}$, $S_2 = \{$ FR$\}$ and $S_3 = \{$ CNTK $\}$, and explain the difference in performance due to gate learning. $S_2$ and $S_3$ have no gate learning. However, $S_3$ due to its infinite width has better averaging resulting in a well formed kernel and hence performs better than $S_2$ which is a finite width. Thus, the difference between $S_2$ and $S_3$ can be attributed to finite versus infinite width. Both $S_1$ and $S_2$ are finite width, and hence, conventional feature learning happens in both $S_1$ and $S_2$, but, $S_1$ with gate learning is better (77.5% or above in CIFAR-10) than $S_2$ (67% in CIFAR-10) with no gate learning. Thus neither finite width, nor the conventional feature learning explain the difference between $S_1$ and $S_2$. Thus, 'gate learning' discriminates the regimes $S_1$, $S_2$ and $S_3$ better than the conventional feature learning view.

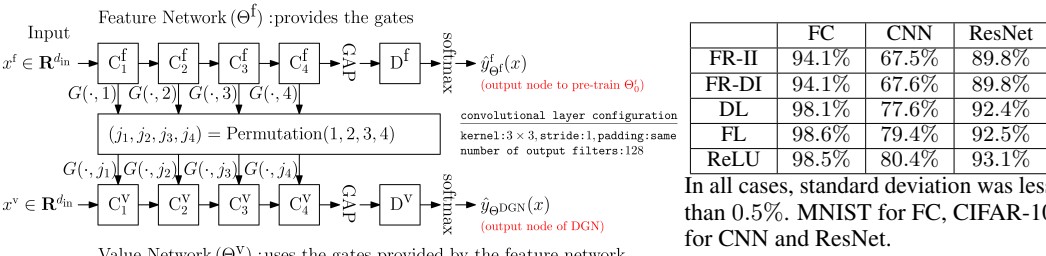

|  | FC | CNN | ResNet |
|---|---|---|---|
| FR-II | 94.1% | 67.5% | 89.8% |
| FR-DI | 94.1% | 67.6% | 89.8% |
| DL | 98.1% | 77.6% | 92.4% |
| FL | 98.6% | 79.4% | 92.5% |
| ReLU | 98.5% | 80.4% | 93.1% |

In all cases, standard deviation was less than 0.5%. MNIST for FC, CIFAR-10 for CNN and ResNet.

Figure 4: $\mathrm{C}_i^{\mathrm{f}}, \mathrm{C}_i^{\mathrm{v}}, i \in [4]$ are the convolutional layers, which are followed by *global-average-pooling* (GAP) layer then by a dense layer ($\mathrm{D}^{\mathrm{f}}/\mathrm{D}^{\mathrm{v}}$), and a softmax layer to produce the final logits.

(Left) Standard ReLU DNN and (Right) DGN with Fixed Learnt Gates from model on the left with $j_1, j_2, j_3, j_4 = 4, 3, 2, 1$ and $x^v = \mathbf{rand-tensor}$

For each model, input is shown first and then starting from the first layer, the first 2 filters of each of the 4 layers are shown.

Figure 5: Hidden layer outputs for a fixed random input to the value network of DGN with permuted gating.

5. *Permutation and Input Invariance:* The performance (in all the $4$ regimes) is robust to 'all-ones' inputs. Note that in the 'all-ones' case, the input information affects the models only via the gates. Here, all the entries of the input Gram matrix are identical, and the NPK depends only on $\Lambda$, which is the measure of sub-network active simultaneously for the various input pairs. The performance (in all the $4$ regimes) is also robust to permutation of the layers. This can be attributed to the product $\Pi_{l=1}^{(d-1)} H_{l,\Theta}^{\text{lyr}}$ of the layer level base kernels being order invariant.

6. *Visualisation:* Figure 5 compares the hidden layer outputs of a standard DNN with ReLU with $4$ layers, and that of a DGN which copies the gates from the standard DNN, but, reverses the gating masks when applying to the value network. Also, the value network of the DGN was provided with a fixed random input (as shown in Figure 5). Both the models achieved about $80\%$ test accuracy, an otherwise surprising outcome, yet, as per the theory developed in this paper, a random input to the value network should not have much effect on performance, and this experiment confirms the same.

## 7 RELATED AND FUTURE WORK

Our paper extended the work of Lakshminarayanan and Singh (2020) to CNN and ResNet. Further, we pointed out the *composite* nature of the underlying kernel. Experiments with permuted masks and constant inputs are also significant and novel evidences, which to our knowledge are first of their kind in literature. Gated linearity was studied recently by Fiat et al. (2019), however, they considered only single layered gated networks.Jacot et al. (2018); Arora et al. (2019); Cao and Gu (2019); Jacot et al. (2019); Du et al. (2018) have all used the NTK framework to understand questions related to optimisation and/or generalisation in DNNs. We now discuss the future work below.

1. *Base Kernel:* At randomised initialisation, for each $l$, $\frac{H_{l,\Theta_0}^{\text{lyr}}(s,s')}{w}$ is the fraction of gates that are simultaneously active for input examples $s, s'$, which in the limit of infinite width is equal to $\left(\frac{1}{2} - \text{angle}(z_{x_s}(\cdot,l), z_{x_{s'}}(\cdot,l))\right)$ (Xie et al., 2017). Further, due to the property of ReLU to pass only positive components, we conjecture that the pairwise angle between input examples measured at the hidden layer outputs is a decreasing function of depth and as $l \to \infty$, $\frac{H_{l,\Theta_0}^{\text{lyr}}(s,s')}{w} \to \frac{1}{2}, \forall s, s' \in [n]$. We reserve a formal statement on the behaviour of $H_{l,\Theta_0}^{\text{lyr}}$ for the future.

2. *Multiple Kernel Learning* (Gönen and Alpaydın, 2011; Bach et al., 2004; Sonnenburg et al., 2006; Cortes et al., 2009) is the name given to a class of methods that learn a linear or non-linear combination of one or many base kernels. For instance, Cortes et al. (2009) consider polynomial combinations of base kernels, which also has a 'sum of products' form. Our experiments do indicate that the learning in the gates (and hence the underlying base kernels) has a significant impact. Understanding $K^{\text{f}}$ (Proposition 5.1) might be a way to establish the extent and nature of kernel learning in deep learning. It is also interesting to check if in ResNet the kernels of its sub-FC-DNNs are combined optimally.

## 8 CONCLUSION

We attributed the success of deep learning to the following two key ingredients: (i) a composite kernel with gates as fundamental building blocks and (ii) allowing the gates to learn/adapt during training. We justified our claims theoretically as well as experimentally. This work along with that of Lakshminarayanan and Singh (2020) provides a paradigm shift in understanding deep learning. Here, gates play a central role. Each gate is related to a hyper-plane, and gates together form layer level binary features whose kernels are the base kernels. Laying out these binary features depth-wise gives rise to a product of the base kernels. The skip connections gives a 'sum of product' structure, and convolution with pooling gives rotation invariance. The learning in the gates further enhances the generalisation capabilities of the models.

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
