# OpenReview forum: "Deep Learning Is Composite Kernel Learning"
_ICLR.cc/2021/Conference — Reject_

### Official Review · AnonReviewer2 · 2020-10-27
**Comments to "Deep Learning Is Composite Kernel Learning "**

**Rating:** 6
**Confidence:** 1

**Review:**

#### General Comments
This paper establishes close relationship between CNN and FC-DNN with a composite kernel method. Specially,  this paper shows that architectural choices such as convolutional layers with pooling, skip
connections, make deep learning a composite kernel learning method, where the
kernel is a (architecture dependent) composition of base kernels. This interestingly indicates that  standard deep networks have in-built structural properties that may explain their success before training them.
Moreover, this paper develops neural path framework to characterize the role of gates/ masks in FC-DNN.
#### Specific Comments
(1) It would be more interesting if some superiority of deep learning relative to kernel methods can be provided.
(2) Lakshminarayanan and Singh (2020) has developed a neural path framework in the NTK regime.  Are there additional challenges when establishing   these similar conclusions for DNNs with Relu activation?

---

> ### Author Response · Authors · 2020-11-16
> **Response to AnonReviewer2**
>
> Thanks for your comments.
> 1. Prior works show that performance of standard finite width deep networks (our focus) > infinite width neural tangent kernel counterparts > infinite width GP counterparts.
> \begin{array}{|lc|}\hline
> \text{Work}&\text{CIFAR-10 (Test Acc.)}\newline\hline
> \text{[Lee et al.,  2018]}& \text{55.66%}\newline
> \text{Fully Connected GP}&\text{(45k points)}\newline\hline
> \text{[Novak et al.,  2019]}& \text{67.14%}\newline
> \text{Convolutional GP}&\newline\hline
> &&\newline
> \text{[Arora et al., 2019]}& \text{77.43%}\newline
> \text{Conv. Neural Tangent Kernel}&\newline
> \text{(CNTK)}&\newline\hline
> \text{[Lakshminarayanan}&\text{67.1% (random gates)}\newline
> \text{and. Singh, 2020]}&\text{79.68% (learnt gates)}\newline
> &\text{80.32% (standard CNN)}\newline\hline
> \end{array}
>
> 2. In DNN with ReLU, the assumption that $\Theta^v_0$ is statistically independent of $\Theta^f_0$ does not hold. However, experiments show that statistically decoupling $\Theta^v_0$ and  $\Theta^f_0$ does not degrade the test accuracy.

---

### Official Review · AnonReviewer4 · 2020-10-27
**Interesting work, bringing a new view on the neural tangent kernel**

**Rating:** 6
**Confidence:** 2

**Review:**

This paper builds on recent work characterising deep neural networks in terms of Neural Tangent Kernels and Neural Path Features. Over the past few years, a number of papers have developed the theory of Neural Tangent Kernels, which can be used to interpret infinite width deep neural networks in the context of a particular type of kernel. A recent paper (Lakshminarayanan and Singh, NeurIPS 2020) provided a new perspective on Neural Tangent Kernels for Gated Neural Networks, by decomposing the network into independent paths. For a fixed set of network weights, we can consider each path to give rise to a feature, corresponding to whether this path is active (i.e., is not switched off by one of the gates on the path). Then, the output of the neural network can be viewed as a weighted sum of active paths, equivalently the dot product of the neural path feature vector and a neural path value vector. Lakshminarayanan and Singh showed that under certain assumptions, a kernel defined in terms of the neural path feature is approximately equal to the neural tangent kernel (up to a constant). Specifically, they show that the value of the neural tangent kernel matrix tends to a constant multiple of the neural path kernel matrix as the width of the network goes to infinity. This suggests that the key component in a deep neural network with RELU activations is the gating structure, which defines active subnetworks, as opposed to the values.

As far as I can see, this work extends the analysis of (Lakshminarayanan and Singh, NeurIPS 2020) in two ways. Firstly, the analysis is extended to certain ResNet and Convolutional architectures, showing that in both of these cases we can relate the neural tangent kernel matrix to the neural path kernel matrix using a result analogous to Theorem 5.1 in (Lakshminarayanan and Singh, NeurIPS 2020). Secondly, they provide an interpretation of the neural path kernel as a composite kernel composed of layer-wise kernels, giving rise to the title of the paper.

I was not very familiar with the work on neural tangent kernels and encountered (Lakshminarayanan and Singh, NeurIPS 2020) for the first time when reviewing this paper. As such, there were things which I didn't fully understand and may have misunderstood in my review.

I have one question regarding the theoretical results in both papers. Theorem 5.1 (in both papers) relates the neural path kernel to the neural tangent kernel by showing that the neural tangent kernel for a network in which the gates have been fixed tends to a constant multiple of the neural path kernel as the width of the nerwork goes to infinity. This felt counter-intuitive to me at first reading, as fixing the gates and growing the width to infinity seem to be mutually exclusive. Is it correct to interpret the result as follows? For any fixed gating structure, there is a relationship between the neural path kernel matrix and the neural tangent kernel matrix for a network with that gating structure (i.e., one in which we are only learning the neural path values). As we allow the width go to infinity this relationship tends to one of equality (up to a constant multiple).

I also have a number of questions regarding the empirical results in this paper.
1. In the experiments where we have fixed the weights, is the model learning parameters in a network in which the gating structure is fixed or is it learning the neural path value vector as part of a linear model?
2. The discussion mentions performance when we fixed the input gram matrix to be a constant in the definition of the neural path kernel (and hence define the neural path kernel in terms of the gating structure only), but does not include numerical results for this case. I understand that this may be due to a desire to keep the paper within the recommended 8 pages, but don't see why these results could not have been added in the appendix.
3. The discussion mentions performance when we permute the layers of the model and claims this is robust to permutation of the layers, but but does not include numerical results for this case. As above, I understand that this may be due to a desire to keep the paper within the recommended 8 pages, but don't see why these results could not have been added in the appendix. Moreover, I wasn't not sure what being robust to permutation of the layers means for the case where we are learning the both components of the DGN. Is this claiming that the results do not change if we permute the layers after training in the DL regime?

Additional comments
1. LS2020 is used in several places to refer to (Lakshminarayanan and Singh, NeurIPS 2020), the recent paper on which this builds. These should be corrected to match the format of the other citations.
2. Table 1 contains the information flow for the FCNN. Arguably, this is the simplest of the three architectures and an illustration of the information flow for a CNN could be more useful here. At the very least, the authors should direct the reader to Appendix A, where the CNN is described in more detail.
3. The authors refer to this work and the previous work of (Lakshminarayanan and Singh, NeurIPS 2020) as a "paradigm shift in understanding deep learning". While this work seems to be an interesting and promising line of research, I think it is fair to say that we will need to wait to see if it really does provide a paradigm shift in how we understand deep learning.

In gener`al, I think the neural tangent kernel is an interesting and promising line of research in the study of deep neural networks. The recent work of Lakshminarayanan and Singh (NeurIPS 2020) seems to add to this and this paper provides a relevant follow-up up that and as such is likely to be of interest to the ICLR community. However, I was did not check the proofs in the appendix or the appendix of (Lakshminarayanan and Singh, NeurIPS 2020), on which the results in this paper depend, and hope another reviewer more familiar with this line of work was able to do so.

---

> ### Author Response · Authors · 2020-11-16
> **Response to AnonReviewer4**
>
> Thanks for the detailed comments. We address the main points below, and we will fix the cosmetic issues pointed out in the final draft.
>
> 1. Width going to infinity: It is a correct observation “as the width goes to infinity the relationship tends to equality (up to a constant value)”.
>
> 2. We are not training a linear model with the neural path features, but the model learning parameters in a network (this is the value network) in which the gating structure (provided by the feature network) is fixed.
>
> 3. Results of layer permutation and 'all-ones' input:
> The values in the table in Fig 4 are already averaged over the combinatorially many models (24 layer permutations and 2 input configurations =48 models). We missed out mentioning this detail in the paper and thanks for pointing it out. Since the deviation was less than $0.5\%$ we did not present them (as mentioned below the table in Fig 4).
>
> 4. Robustness in DL regime:
> In the DL regime, we permute the masks (when using them in the value network) during training itself, this gives us 24 models. For each of these 24 models, the input can  be set to be the image or `all-ones', during training. And robustness here means, that the test accuracy is more or less the same (within $0.5\%$. deviation) for all these 48 different models.  No, we do not claim that the results do not change if we permute the layers after training in the DL regime. The fact that we can permute the masks after training and retrain the NPV to recover performance is established in the FL regime.

---

### Official Review · AnonReviewer3 · 2020-10-28
**Paper proposes an extension of the neural path framework to include composite kernel (sum, product-sum and CNN pooling) learning and learnt gates to show for infinite kernel width gates are important than weights.**

**Rating:** 8
**Confidence:** 2

**Review:**

Paper proposes and extension of neural path framework to include composite kernels which comprise of a) FC networks (Hadamard product of gram matrices), b) residual networks (sum of products of base kernels), and c) CNN max-pooling layer. Furthermore, they also include learnt gates instead of static initialized random gates and show learnt gates perform better.

Paper is well written with main technical contribution being theorem 5.1 which shows for infinite width case $w \rightarrow \infty $ the NTK is independent of the weights. It also presents experimental result on MNIST and CIFAR for four proposes regimes of (Definition 5.1) that models are robust to combinatorial variations in layers and inputs.  This results in novel makes an important theoretical contribution towards understanding of why DNN with composite kernels perform well in practice.

---

> ### Author Response · Authors · 2020-11-16
> **Response to AnonReviewer3**
>
> Thanks for appreciating our contributions.

---

### Official Review · AnonReviewer1 · 2020-10-30
**Ok paper, but needs better exposition and model details**

**Rating:** 4
**Confidence:** 4

**Review:**

Overview: The authors examine deep learning from the perspective of kernel methods and demonstrate that convolution layers in these architectures can make DNNs a form of composite kernel learning.

Significance: Understanding and interpreting neural networks is an important problem in general; similarly extracting good features key to downstream performance of a neural network. Hence the paper tries to address some important and relevant problems in the field, however, I'm not fully convinced as to whether their procedure is  any more interpretable than existing methods or extracts features optimally.

Quality and Clarity: While the work provides sufficient details to understand prior work and the method itself, the key contribution section of the paper needs more work.

Novelty: There are many works that focus on understanding neural networks and learning features for downstream prediction from the perspective of kernels. The novelty in this work is limited to allowing gating functions to adapt during training, such that the learnt gates can perform better than random gates.

Pros:
1) Paper presents a potential solution to a relevant problem
2) Paper provides good overview of an existing method that form the basis of the approach.

Cons:

1) The most significant weakness of this paper is lack of thorough discussion about what the kernels actually mean in terms of understanding what the neural network is doing. How are these kernels learnt? For this, I think the authors need to make concrete comparisons with methods that are deeply rooted in kernels such as GPs or BNNs. For instance, does using a particular composite kernel structure give you the same predictive performance as when using a GP? Can we directly interpret such models as forms of BNNs or GPs? How does this work compare to more classic work that uses composite kernels in support vector machines?

2) I would have also liked to have seen a better exposition of the interpretability of the method. How does using these composite kernels together compare to other approaches for interpreting deep neural networks like for instance layerwise relevance propagation?

3) There is hardly any reference material which suggests the authors need to include a more thorough description and comparison of related work.

---

> ### Author Response · Authors · 2020-11-16
> **Response to AnonReviewer1**
>
> Thanks for your interesting and useful comments. We clarify that we are not proposing i) a new composite kernel method inspired by/built around deep nets, and ii) a new method to optimally extract features to be used for downstream tasks. Our goal is to understand standard finite width deep networks with ReLU activations, for which, we extend work of [Lakshminarayanan and Singh, 2020].
> 1. Comparison with GPs and SVMs: Prior works show that performance of standard finite width deep networks (our focus) > infinite width neural tangent kernel counterparts > infinite width GP counterparts.  We are not proposing a new composite kernel method, instead we are  showing via analysis that the neural path kernel (NPK) has a composite structure in itself. This is also the reason why we have not compared SVMs with composite kernels.
> \begin{array}{|lc|}\hline
> \text{Work}&\text{CIFAR-10 (Test Acc.)}\newline\hline
> \text{[Lee et al.,  2018]}& \text{55.66%}\newline
> \text{Fully Connected GP}&\text{(45k points)}\newline\hline
> \text{[Novak et al.,  2019]}& \text{67.14%}\newline
> \text{Convolutional GP}&\newline\hline
> &&\newline
> \text{[Arora et al., 2019]}& \text{77.43%}\newline
> \text{Conv. Neural Tangent Kernel}&\newline
> \text{(CNTK)}&\newline\hline
> \text{[Lakshminarayanan}&\text{67.1% (random gates)}\newline
> \text{and. Singh, 2020]}&\text{79.68% (learnt gates)}\newline
> &\text{80.32% (standard CNN)}\newline\hline
> \end{array}
> 2. Meaning and Interpretability of Kernels: [Lakshminarayanan and Singh, 2020] show that most information is stored in the gates, which makes it worthy to analyse the gates in a standalone manner. The NPK is solely related to the gates and sub-networks, unlike GPs related to outputs and neural tangent kernel (NTK) related to gradients. The relation NTK = const x NPK captures the role of gates analytically within the existing NTK framework. Also, NPK  = $\Sigma\odot \Lambda$ helps in interpretability. $\Sigma$ is input Gram matrix, $\odot$ is the Hadamard product, and  $\Lambda$ is physically interpretable since it measures the overlap between active sub-networks.  Prior works rooted in kernels (GPs and NTK) propose and study an infinite width kernel. However, our focus here is to use the insights obtained from the NPK, and instead use the deep gated network (DGN) setup to experiments with different gating regimes in finite width networks (our focus).
>
> 3. Interpreting models as BNNs and GPs: As we are not proposing any new models based on composite kernels, question of interpreting them as BNNs and GPs do not arise in the first place.
>
> 4. Novelty of our work: As mentioned in the paper, the novelties are:
>
>  a) We show that the NPK is composed of base kernels. The base kernels correspond to gating features of individual layers, and each gate is related to the hyperplane given by its incoming edges. $\newline$
> b) Setting $\Sigma=$ matrix of all same entries (by giving `all ones' input to value network) does not degrade test accuracy. This shows that all the useful information is in $\Lambda$, i.e.,  gates/sub-networks.
> c) Our experiments challenge the hierarchical view of feature learning, wherein it is believed that, as one proceeds in depth, lower level to higher level features are learnt in the hidden layers of a deep network.  We show that performance does not degrade if the masks are permuted, i.e., higher layer masks can be used first and lower layer masks in the end.
>
> 5. 'How are Kernels Learnt?' translates to 'How are gates learnt?’. We are not proposing new methods to learn gates. The gates in standard finite width deep nets are learnt by training via off-the-shelf gradient methods.
>
> 6.  Neural path feature is a quantity arising in an analytical framework, and needs no additional procedure, unlike Layer-wise relevance propagation where one needs additional backward passing to compute the relevance of the pixels.
> 7. We added the most relevant references to this work, and we believe the work along with the references is self-contained. However, we will be happy include more references and discuss them in the related work.
>
> References
>
> [1] Jaehoon Lee, Yasaman Bahri, Roman Novak, Samuel S. Schoenholz, Jeffrey Pennington, Jascha Sohl-Dickstein, DEEP NEURAL NETWORKS AS GAUSSIAN PROCESSES, ICLR 2018.
>
> [2] Roman Novak, Lechao Xiao, Jaehoon Lee, Yasaman Bahri, GregYang, Jiri Hron, Daniel A. Abolafia, Jeffrey Pennington, Jascha Sohl-Dickstein, BAYESIAN DEEP CONVOLUTIONAL NETWORKS WITH MANY CHANNELS ARE GAUSSIAN PROCESSES, ICLR 2019.

---

### Decision · Program_Chairs · 2021-01-07
**Final Decision**

**Decision:**

Reject

**Comment:**

This paper provides a new perspective on deep networks by showing that NPK is composed of base kernels and their dependence on the architecture is explicitized. It is further shown that learning the gates can perform better than random gates.

While the paper provides interesting understanding neural networks, it is unclear what practical benefit can be drawn from it. On the architectures considered such as FC, ResNet and CNN (btw, it seems restricted to 1-D), it will be important to show that such insights lead to new models or learning algorithms that improve upon the standard practice in deep learning (or get very close to).  It is debatable whether drawing such a nontrivial insight alone warrants publication at ICLR, while "nontrivial" itself is a subjective judgement.  I understand people differ in their opinions, and the NTK paper has been impactful.  Unfortunately since there are quite a few other papers that are stronger, I have to recommend not accepting this paper to ICLR this time.